# CODE: Contrasting Self-generated Description to Combat Hallucination in Large Multi-modal Models

Junho Kim[*]     Hyun Jun Kim[*]     Yeon Ju Kim     Yong Man Ro[†]

Integrated Vision and Language Lab, KAIST, South Korea
{arkimjh, kimhj709, yeonju7.kim, ymro}@kaist.ac.kr

## Abstract

Large Multi-modal Models (LMMs) have recently demonstrated remarkable abilities in visual context understanding and coherent response generation. However, alongside these advancements, the issue of hallucinations has emerged as a significant challenge, producing erroneous responses that are unrelated to the visual contents. In this paper, we introduce a novel contrastive-based decoding method, COuntering DEscription Contrastive Decoding (CODE), which leverages self-generated descriptions as contrasting references during the decoding phase of LMMs to address hallucination issues. CODE utilizes the comprehensive descriptions from model itself as visual counterpart to correct and improve response alignment with actual visual content. By dynamically adjusting the information flow and distribution of next-token predictions in the LMM's vocabulary, CODE enhances the coherence and informativeness of generated responses. Extensive experiments demonstrate that our method significantly reduces hallucinations and improves cross-modal consistency across various benchmarks and cutting-edge LMMs. Our method provides a simple yet effective decoding strategy that can be integrated to existing LMM frameworks without additional training.

## 1 Introduction

With recent advancements of Large Language Models (LLMs) [18, 4, 54, 11, 46], Large Multi-modal Models (LMMs), sometimes referred as Large Vision-Language Models [24, 63, 32, 31], have been drawn great attention for their natural multi-modal interaction with users through back-and-forth conversations. Leveraging their robust generation capabilities, various pioneering tasks in pre-LMM era such as image captioning [7, 58, 27], visual question answering [1, 2], object detection [17], etc., have been integrated into a single task rather than treated as sub-tasks and achieved significant milestones [21, 59, 44]. However, at the same time, the hallucination issue [49, 67] has become one of the emerging problems when adopting LMMs into real-world applications due to their potential spurious generation in critical areas.

Here, unlike hallucination studies in LLMs [22] mainly focusing on factuality hallucination originated from the language knowledge, the hallucination problem in LMMs refers cross-modal inconsistency between the given visual contents and the generated responses for the user instructions. After the seminal works [13, 42, 68] giving eyes to LLMs to understand visual contents with visual instruction tuning, numerous cutting-edge LMMs [61, 9, 41, 36] actively have been proposed. Albeit the scaling laws following more stronger versatile vision models [37, 29, 33], higher resolution [41, 8], deeper alignment layers [5, 45, 8], larger model sizes, etc., LMMs still suffer from generating responses that seem plausible but are factually incorrect for the given visual contents.

---

[*]Equal contribution. † Corresponding author.
Code is available at https://ivy-lvlm.github.io/CODE

38th Conference on Neural Information Processing Systems (NeurIPS 2024).

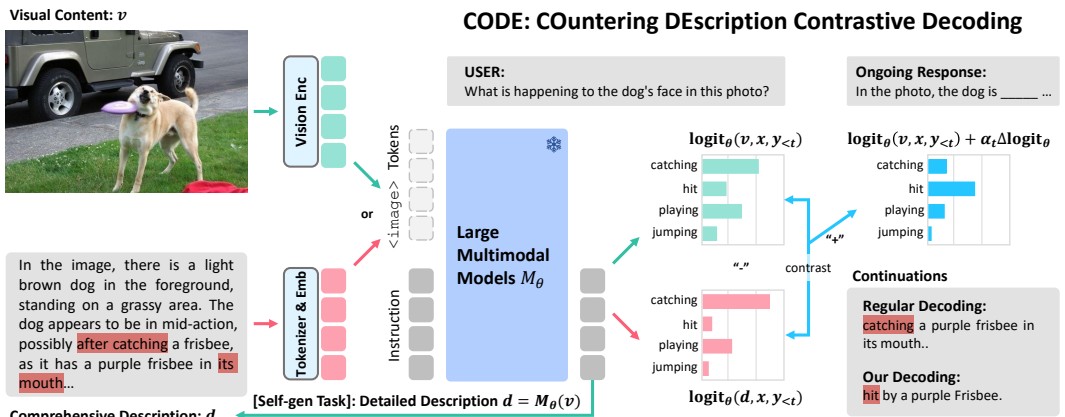

Figure 1: The overall decoding procedure of CODE. After LMMs generate detailed description for the visual content by themselves (see instruction in Appendix. A), the model recursively outputs logits from each $v$ and $d$. By contrasting between two log-likelihoods, CODE produces more contextual and correct responses that match the given visual content suppressing inconsistent words (*catching→hit*).

The origin of LMM hallucination is an intertwined problem for their inherent training paradigm, which involves alignment projection matching during the pre-training, followed by fine-tuning with the limited instruction-following data. Several approaches, aimed for mitigating hallucinatory effects, have addressed the inconsistent issues in the context of data-associated solution [39, 56], scaling model architectures [65], or additional RL-based training [64, 52]. Among them, reactive methods [23, 14, 26] intervene the decoding phase of LMMs' inference and alleviate undesired responses. Motivated by Li *et al.* [34] that have proposed contrastive decoding (CD) method between expert and amateur language models, recent CD-based approaches in LMMs have proposed several ways of contrasting model responses from visual inputs with their counterparts (*e.g.,* visual contamination [30], image-biased models [69], or fine-grained visual information [10]).

Our research question begins with *"How effectively do contemporary LMMs capture visual evidences in their descriptive responses, and what information must be curbed to produce informative and consistent responses?"*. As illustrated in Fig 1 (bottom-left), when asking LMMs to generate a comprehensive description for visual content, the output seemingly generates detailed description effectively, but a closer examination often reveals missed fined-grained information or hallucinatory instances in the responses. By recursively referring these incomplete descriptions generated by the models themselves, we aim to restrict the incorrect information flow during the generation phase and enhance the alignment of the model responses grounded in true visual evidences.

In this paper, we introduce a novel training-free contrastive decoding method, COuntering DEscription Contrastvie Decoding (CODE), designed to use self-generated descriptions as a contrastive reference to mitigate hallucination issues in LMMs. The core idea of our proposed method is on harnessing the self-generated descriptions which possibly encompass both factual evidence and hallucinatory information from visual contents as a look-up reference for response correction. Specifically, within our contrastive framework as illustrated in Fig. 1, the comprehensive descriptions from model itself alternatively propagate to visual input tokens and contrast the discrepancy with the logits from actual visual contents to enhance next-token prediction. In addition, we introduce a dynamic restriction strategy that enables adaptive control of information flow during the auto-regressive decoding phase, taking into account both token-level predictions and their distribution within the vocabulary set.

By conducting extensive experiments and analyses on prevailing cutting-edge LMMs [40, 51, 16, 41, 62, 8], we corroborate the effectiveness of our method in reducing hallucination and enhancing the coherence and informativeness in various benchmarks [35, 42, 52, 53, 57]. Our decoding method can be seamlessly integrated into existing LMMs by simply substituting the image tokens with self-generated descriptions in a training-free manner.

Our contribution can be summarized into three-fold as follows:

- We introduce COuntering DEscription Contrastive Decoding (CODE), a training-free decoding strategy that employs self-generated descriptions to minimize hallucinations in LMMs.

By contrasting logit information from descriptions with actual visual contents, CODE enhances visual consistency and coherence in the model responses.

- Our approach incorporates dynamic restriction strategies within the contrastive decoding phase. It selectively regulates the information flow by adjusting token-level predictions based on their distribution in the vocabulary, thus ensuring more contextual responses.

- We validate the effectiveness of our decoding method across various benchmarks using cutting-edge LMMs. The results demonstrate that CODE significantly reduces hallucination while enhancing the relevance and informativeness in the responses.

## 2 Related Work

**Vision+LLM: Large Multi-modal Models**    After the emergence of large-scaled LLMs [4, 54, 11] that can interact with users with question-answer chat, various of vision+LLM studies— *i.e.,* LMMs, have been proposed to integrate the robust linguistic capability into the visual understanding and reasoning in diverse vision-language task. As earlier works such as LLaVA [42], Instruct-BLIP [13], and MiniGPT-4 [68], which utilized visual instruction tuning, have bridged two modalities of vision and language through fine-tuning with a learnable query, exemplified by Q-Former [31] or projection layer-based alignments [40]. To enhance cross-modal consistency in vision-language representation, recent works have proposed several solutions to address the underlying weaknesses in both modalities: (i) utilizing higher-resolution visual inputs [41, 9], (ii) deploying Mixture-of-Expert (MoE) concepts integrating versatile vision models [37, 29, 33], (iii) improving weak alignment interface [5, 45, 8], or (iv) adopting larger LLMs to scale up the language model prior [62].

**Hallucination Issue, Harming Cross-modal Consistency**    Despite of the endeavor developments of LMMs, they cannot be free from cross-modal inconsistency between the visual contents and their generated responses, so-called hallucination [49]. This not only leads to performance degradation but also provokes an over-reliance issue, resulting in incorrect model responses that are not grounded in true visual evidence. This critical concern regarding response trustworthiness and model reliability hinders the adoption of LMMs in real-world applications. To mitigate the hallucination problem, diverse works have been proposed employing additional training on curated datasets [52, 55] or reinforcement learning under feedback systems [64, 66]. Among them, by intervening during the response generation, decoding-based approaches [34, 12] are introduced to encourage models to represent more precise responses. We refer to readers for more comprehensive survey papers [43, 3] addressing hallucination in LMMs. Our work is in line with CD-based approaches that utilize logit discrepancy from counterpart outputs to enhance coherence. Unlike the previous works [30, 23, 10] that focus on twisting visual information, we utilize self-generated description as contrasting visual counterpart and correct hallucinatory responses based on the model understanding.

## 3 Proposed Method

**Problem Setup and Preliminaries.**    Let $M_\theta$ denote a vision-language LMM parameterized by $\theta$ that auto-regressively generates responses for the given visual contents $v$ and input textual query $x$. Then the model maps the logit distribution to the next token prediction output $y_t \in \mathbb{R}^{|\mathcal{V}|}$ at time step $t$ in the vocabulary set $\mathcal{V}$ such that $y_t \sim p_\theta(y_t|v, x, y_{<t}) \propto \text{logit}_\theta(y_t|v, x, y_{<t})$, where $y_{<t}$ indicates all previously generated tokens. During the response generation, we can deploy several decoding strategies to choose the next word using either deterministic search (*e.g.,* greedy, beam search) or stochastic sampling (*e.g.,* top-k, Nucleus search [20]).

After the seminal works [34, 12] in natural language processing have introduced Contrastive Decoding (CD) mechanism, which considering information disparities between expert and amateur models for more coherence and informativeness, various works have deployed this strategy into LMMs by twisting visual contents [30] or model information [69] for the contrastive approach. The next-token probability $p_{\text{cd}}$ from CD can be generally formulated as follows:

$$p_{\text{cd}}(y_t \mid y_{<t}) = \text{Softmax}\left[(1 + \alpha)\text{logit}_\theta(y_t \mid v, x, y_{<t}) - \alpha\text{logit}_{\hat{\theta}}(y_t \mid \bar{v}, x, y_{<t})\right], \quad (1)$$

where $\bar{v}$ and $\hat{\theta}$ indicates visual counterparts and sub-optimal amateur model, respectively— note that $\theta{=}\hat{\theta}$ can be regarded as self-correction. Intuitively, the objective of CD is amplifying model outputs

by reflecting information deviation between top candidate log-probabilities. Therefore, the selection of logit counterparts for referring is the key challenge for high-quality and consistent responses during the contrastive decoding frameworks.

## 3.1 Comprehensive Image Description as Visual Counterpart

As the visual counterpart, we deploy comprehensive image descriptions generated from the model as contrasting reference, which are inevitably less informative than the visual contents themselves. Our motivation is on the innate difference of information density between vision and language [19]. While vision information exhibits relatively less redundancy for spatial signals— *e.g.,* human can visually recognize objects with a few masked patches on images, languages contain high-entropy information, resulting in spans that are more semantic and information-dense than visual signals. That is, it is difficult to infer blanked-out words— *e.g.,* "I went to the store to buy some ____". Building upon the property of each modality, we first delve into the self-generated model responses with specific instruction for the visual contents to elicit the encompassed visual evidences in the representation space and assess its potential subject role as a visual counterpart for contrastive decoding.

As illustrated in Fig. 1, we input a query instruction into the model to generate a comprehensive visual description for the given visual content (please see the detailed instruction $x_0$ for the self-generated description in Appendix. A). Then, we exploit the generated description as recursive visual inputs, replacing the position of image tokens in the model input sequence (*e.g.,* `<image>` token in LLaVA series [42, 40, 41]). Ideally, if the model sufficiently covers whole visual evidences to answer any vision-related questions, the generated response should provide competent answers with solely utilizing the description-only embeddings as an alternative of visual embeddings. However, as shown in Fig 2, it is obvious that the results from description-only show sub-optimal performance to answer questions due to insufficient information in capturing visual evidences. Consequently, we use the comprehensive description for the visual contents, which is generated by model itself— but partially incorrect or hallucinatory to capture visual evidences, as a contrasting visual counterpart to enhance response coherence during the decoding phase, which also in line with the amateur model selection philosophy of contrastive decoding [34].

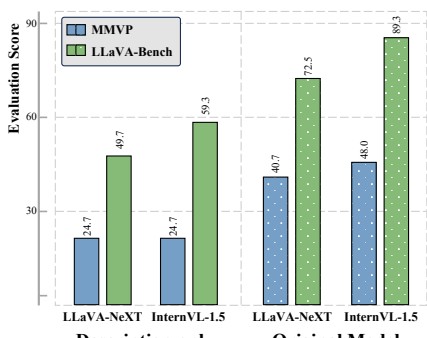

Figure 2: The comparison is based on two benchmarks (MMVP [53]: multiple choice / LLaVA-Bench [42]: description-level). The plain and dotted bars indicate the results for the models that use self-generated descriptions as visual input replacements and original model with actual visual contents, respectively.

## 3.2 COuntering DEscription Contrastive Decoding

Based on our analysis in sec. 3.1, we can obtain a pair of the visual content and its comprehensive description $(v, d)$, such that $d$ corresponds to $M_\theta(y|v, x_0)$. By contrasting the logit variation between the paired information into the model response generation, we can formulate the next-word prediction using our proposed method, COuntering DEscription Contrastive Decoding (CODE):

$$p_{\text{code}}(y_t \mid y_{<t}) = \text{Softmax}\left[(1 + \alpha_t)\text{logit}_\theta(y_t \mid v, x, y_{<t}) - \alpha_t\text{logit}_\theta(y_t \mid d, x, y_{<t})\right]. \quad (2)$$

Here, unlike the previous approaches [34, 30, 12] that restrict the logit variations with fixed $\alpha$ value as in Eqn. 1, we present a dynamic restriction $\alpha_t$ for the logit variations by comparing the information between visual contents and their comprehensive descriptions. Revisiting the role of $\alpha$, it determines whether to promote or curb information from logit variation, thus directly influencing next-token generation— higher value results in more aggressive adjustment for the variations. However, when confronting that both $v$ and $d$ yield similar logit score on the correct token, the variation gets closer to zero, thereby the next-token prediction can be unexpectedly reversed if other tokens get rewarded than the correct token with a fixed $\alpha$ on a token-by-token basis. Although this aligns with the initial intent of CD, a more robust selector is necessary to effectively restrict the logit information flow.

Accordingly, our method predicts next-token not only at the individual token-level but also considering its distribution across the entire vocabulary set, enabling dynamic control of the information flow.

To measure the relative entropy between the token distributions from visual contents $P_t^v$ and its comprehensive description $P_t^d$ at time step $t$, we deploy Bounded Divergence ($\mathcal{D}_{bd}$) [6], which is a type of statistical distance that ensures symmetric and bounded measure:

$$\mathcal{D}_{bd}(P\|Q) = \frac{1}{2}\sum_{i=1}^{n}(p_i + q_i)\log_2(|p_i - q_i|^k + 1), \qquad (3)$$

where $\mathcal{D}_{bd}(P\|Q) \geq 0$ and equals 0, if and only if $p=q$, and $k$ denotes a smoothing parameter. Here, the upper-bound of the divergence apparently exists, such that $\mathcal{D}_{bd}(P\|Q) \leq \sum_{i=1}^{n} p_i \log_2 2 = 1$, due to the following condition $|p_i - q_i| \leq 1$.

We define the dynamic restriction $\alpha_t$ as $1 - \mathcal{D}_{bd}(P_t^v\|P_t^d)$, where it enables a token-wise feedback control that adjusts the information weighting with respect to the closeness of the two distributions. The major role of the restriction term is maintaining a balance in the logit variation for the observed prediction disparities between the $v$ and $d$ distributions. That is, when the distributions are close enough (*i.e.,* $P_t^v \approx P_t^d$), the value of $\mathcal{D}_{bd}(P_t^v\|P_t^d)$ approaches zero, indicating minimal divergence. That is, $\alpha_t$ approaches 1, allowing for higher amplification of logit variations in predicting the next-token outputs. This adjustment reflects the increased reliability of predictions when the two distributions from $v$ and $d$ are closely aligned. On the other hand, for the dissimilar distributions, $\alpha_t$ decreases towards zero, compelling the model to restrict information flow from the variation. This reduction limits the potential for introducing erroneous or less probable predictions by focusing more on visual information, thereby maintaining coherence in the output when the model's understanding of the visual content significantly deviates from its textual description.

### 3.3 Adaptive Information Constraint

One major challenge in contrastive-based decoding is the scenario where implausible tokens are rewarded, even when predictions are made with low confidence. This issue can also arise in our method, particularly when token distributions derived from textual descriptions provide more confidence than the visual content, ironically undermining the most predictive tokens. To address it, Li *et al.* [34] have introduced an adaptive plausibility constraint, which filters out less plausible tokens by truncating them based on the maximum token confidence from the expert model. While this approach simply penalizes false positive tokens in the candidate pool, it may also have unintended side effects by prematurely applying a cutoff threshold to lower-confidence tokens. Specifically, early threshold settings can sometimes eliminate the possibility of identifying correct token predictions, which might otherwise be dismissed in a pool considered to contain mostly false negatives.

Improving the previous constraint [34, 30], we present adaptive information constraint ($\mathcal{V}_{head}$) designed to dynamically retain tokens that may be informative despite their lower confidence. By comparing prediction distributions between $P_t^v$ and $P_t^d$, we filter out less relevant tokens from the candidate pool as follows:

$$\mathcal{V}_{head}(y_{<t}) = \{y_t \in \mathcal{V} : p_\theta(y_t \mid v, x, y_{<t}) \geq \beta_t \max_w p_\theta(w \mid v, x, y_{<t})\}, \qquad (4)$$

where $\beta_t$ dynamically regulate the token candidate pool utilizing the divergence term in Eqn. 3, defined as $\beta_t = \mathcal{D}_{bd}(P_t^v\|P_t^d)$. This strategy can expand the token searching pool when the next-token prediction, derived from both visual content and comprehensive description, shows a similar distribution yet uncertainty in selecting the candidate token (*i.e.,* false negatives). Finally, we only consider the next-token prediction within $\mathcal{V}_{head}(y_{<t})$, and for the tokens satisfying $y_t \notin \mathcal{V}_{head}(y_{<t})$, we set their logits to $-\infty$ to filter out from the candidate pool. Please see comprehensive Algorithm. 1 in Appendix. B.

## 4 Experiments

**Experimental Setup** To validate the efficacy of our method over various LMM families and sizes, we implemented our method on contemporary LMMs: LLaVA-1.5 (13B) [40], Emu2-Chat (14B) [51], InternLM-XComposer2 (7B) [16], LLaVA-NeXT (34B) [41], Yi-VL (34B) [62], and InternVL 1.5 (26B) [8]. We compared our method with five baseline decoding strategies. For the regular decoding strategies, we used greedy decoding, Nucleus sampling [20], and beam search decoding. Additionally, we selected OPERA [23] and VCD [30] for contrastive decoding method, which designed to mitigate

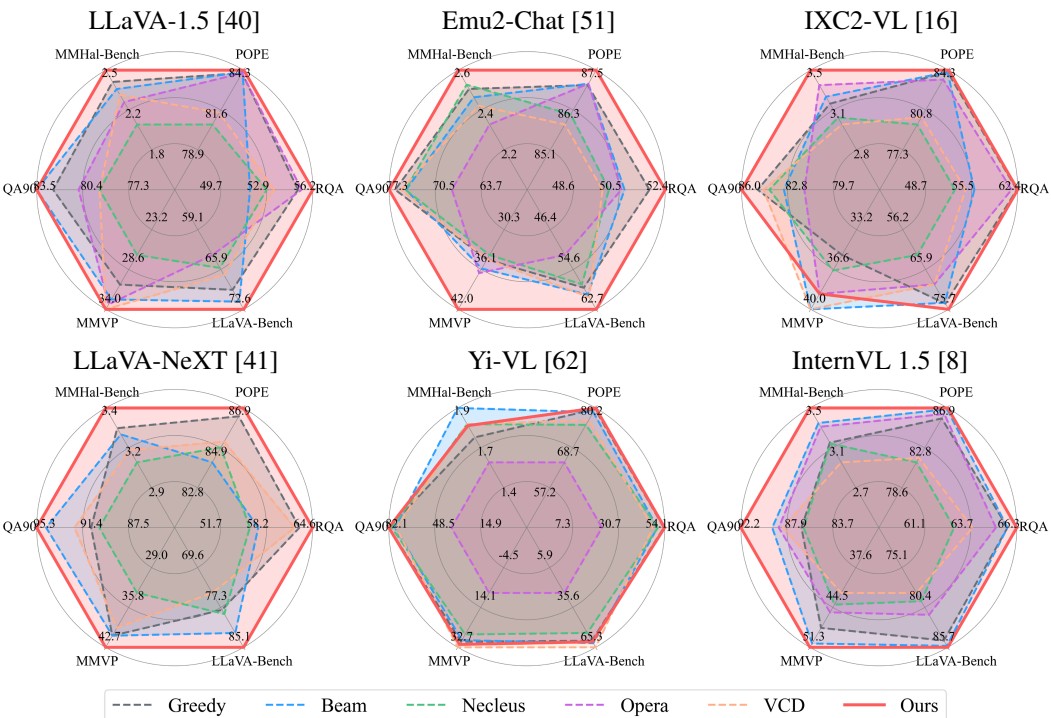

Figure 3: Overview of experimental results on 6 baseline LMMs, 6 decoding method, and 6 hallucination benchmarks in spider chart format.

hallucinations with contrastive frameworks. We used the default parameter settings for all methods, where top-p value $0.95$ and temperature $1.0$ for Nucleus sampling, the number of window size for searching is 5 (*i.e.*, num-beams 5) for both beam search decoding and OPERA, and CD-$\alpha = 1$, CD-$\beta = 0.1$ for VCD, and $k = 0.3$ for our method. Note that OPERA inference requires too much memory especially for LLaVA-NeXT (34B), so that we excluded OPERA results for this model.

**Benchmarks and Evaluation Metrics** The benchmarks for evaluating hallucinations in LMMs can be broadly categorized into discriminative and generative streams. The discriminative type assesses hallucinations by evaluating the predicted answer among given options (*e.g.,* multiple choice or yes/no question), while generative benchmarks typically employ more advanced language models (*e.g.,* GPT-aided evaluation) to rate the subject model descriptions. Within this taxonomy, we carefully select 6 benchmarks to test baselines. Please see Appendix. C for benchmark details.

As discriminative benchmarks, we utilize mainly three datasets for detailed evaluation. Specifically, **POPE** [35] is a commonly used benchmark for detecting object hallucination by converting object annotations sourced from MSCOCO [7]. Under the three different subsets: random, popular, and adversarial, the metric for POPE measures binary classification performance for simple yes/no questions. **MMVP** [53] aims to evaluate the understanding of visual details for 9 different visual patterns using paired classification accuracy. Due to its evaluation design, which involves comparing two similar CLIP-blind image pairs, MMVP requires LMMs to capture subtle visual differences. **RealworldQA** [57] is the most recent dataset tailored to assess the capability of LMMs in basic real-world spatial understanding, using the accuracy metric within multiple-choice questions.

We use three benchmarks for generative benchmarks, extending the evaluation scope to include open-ended captioning tasks beyond merely assessing classification within given answer options. Generally, ChatGPT [46] is used to score the quality of the model-generated sentences. The metric for both **LLaVA-QA90** [42] and **LLaVA-Bench (In-the-Wild)** [42] is score ratio, where model responses rated from GPT-4 [47] are divided by GPT-4 answers such that $\sum |\text{model-score}| / \sum |\text{GT-score}|$, where all scores are rated by GPT-4. It has three types of questions: conversation, detailed description, and complex reasoning. **MMHal-Bench** [52] evaluates the degree of hallucination for the 8 various question types: object attribute, adversarial object, comparison, counting, spatial relation,

Table 1: The hallucination evaluation results for the discriminative benchmarks [35, 53]. Each emoji in MMVP colum cell indicates 9 different visual patterns (details in Appendix. C).

| Model | #Param | Method | POPE | | | MMVP | | | | | | | | | |
|---|---|---|---|---|---|---|---|---|---|---|---|---|---|---|---|
| | | | Acc↑ | Prec | F1↑ | 🧭 | 🔍 | 🔄 | ↕ | 💡 | 🌐 | ⚙ | A | 📷 | Avg↑ |
| LLaVA-1.5 | 13B | Greedy | 84.07 | 90.88 | 82.62 | 30.77 | 27.27 | 0.00 | 12.50 | 10.00 | 53.33 | 16.67 | 50.00 | 40.00 | 30.67 |
| | | Beam | 84.13 | 90.89 | 82.70 | 19.23 | 27.27 | 11.11 | 25.00 | 10.00 | 60.00 | 16.67 | 70.00 | 35.00 | 32.67 |
| | | Nucleus | 80.60 | 84.46 | 79.45 | 26.92 | 27.27 | 22.22 | 12.50 | 20.00 | 33.33 | 0.00 | 60.00 | 20.00 | 26.67 |
| | | Opera | 84.07 | 91.01 | 82.59 | 42.31 | 36.36 | 11.11 | 25.00 | 10.00 | 56.67 | 16.67 | 70.00 | 35.00 | 33.33 |
| | | VCD | 81.40 | 85.00 | 80.39 | 34.62 | 18.18 | 22.23 | 37.50 | 50.00 | 43.33 | 33.33 | 40.00 | 35.00 | **34.00** |
| | | Ours | **84.27** | 90.99 | **82.86** | 19.23 | 31.82 | 11.11 | 25.00 | 20.00 | 53.33 | 16.67 | 80.00 | 40.00 | **34.00** |
| Emu2-Chat | 14B | Greedy | 87.03 | 94.48 | 85.85 | 30.77 | 36.36 | 11.11 | 12.50 | 20.00 | 63.33 | 16.67 | 70.00 | 20.00 | 34.67 |
| | | Beam | 87.07 | 93.99 | 85.96 | 38.46 | 27.27 | 11.11 | 25.00 | 20.00 | 66.67 | 16.67 | 60.00 | 25.00 | 36.00 |
| | | Nucleus | 86.17 | 92.22 | 85.10 | 38.46 | 31.82 | 16.67 | 0.00 | 40.00 | 43.33 | 33.33 | 60.00 | 30.00 | 34.00 |
| | | Opera | 87.07 | 94.06 | 85.95 | 38.46 | 31.82 | 11.11 | 25.00 | 20.00 | 63.33 | 16.67 | 70.00 | 25.00 | 36.67 |
| | | VCD | 85.87 | 91.58 | 84.82 | 34.62 | 36.36 | 7.78 | 0.00 | 0.00 | 56.67 | 0.00 | 70.00 | 20.00 | 34.67 |
| | | Ours | **87.47** | 92.77 | **86.64** | 42.31 | 40.91 | 7.78 | 0.00 | 30.00 | 63.33 | 50.00 | 70.00 | 30.00 | **42.00** |
| IXC2-VL | 7B | Greedy | 84.13 | 83.12 | **84.37** | 34.62 | 31.82 | 11.11 | 37.50 | 50.00 | 43.33 | 16.67 | 70.00 | 30.00 | 35.33 |
| | | Beam | 84.13 | 83.12 | **84.37** | 34.62 | 27.27 | 11.11 | 37.50 | 50.00 | 6.33 | 33.33 | 70.00 | 35.00 | **40.00** |
| | | Nucleus | 79.53 | 77.41 | 80.30 | 34.62 | 27.27 | 16.67 | 62.50 | 30.00 | 53.33 | 0.00 | 80.00 | 25.00 | 36.67 |
| | | Opera | 83.47 | 82.85 | 83.62 | 42.31 | 31.82 | 16.67 | 37.50 | 50.00 | 53.33 | 16.67 | 70.00 | 25.00 | 38.67 |
| | | VCD | 80.17 | 78.23 | 80.83 | 34.62 | 31.82 | 38.89 | 50.00 | 40.00 | 40.00 | 16.67 | 90.00 | 35.00 | **40.00** |
| | | Ours | **84.30** | 86.26 | 83.86 | 34.62 | 27.27 | 11.11 | 37.50 | 6.00 | 56.67 | 16.67 | 70.00 | 35.00 | 38.67 |
| LLaVA-NeXT | 34B | Greedy | 86.50 | 83.86 | 87.01 | 38.46 | 40.91 | 16.67 | 37.50 | 30.00 | 60.00 | 0.00 | 80.00 | 35.00 | 40.67 |
| | | Beam | 84.13 | 90.89 | 82.70 | 38.46 | 31.81 | 22.22 | 37.50 | 50.00 | 60.00 | 0.00 | 80.00 | 30.00 | 40.67 |
| | | Nucleus | 84.90 | 82.78 | 85.37 | 34.62 | 22.73 | 27.78 | 25.00 | 20.00 | 43.33 | 0.00 | 50.00 | 45.00 | 33.33 |
| | | Opera | - | - | - | - | - | - | - | - | - | - | - | - | - |
| | | VCD | 85.20 | 83.00 | 85.68 | 42.31 | 22.73 | 22.22 | 37.50 | 50.00 | 46.67 | 16.67 | 80.00 | 40.00 | 39.33 |
| | | Ours | **86.93** | 83.82 | **87.51** | 34.62 | 36.36 | 33.33 | 25.00 | 50.00 | 70.00 | 0.00 | 70.00 | 30.00 | **42.67** |
| Yi-VL | 34B | Greedy | 79.83 | 87.64 | 77.50 | 23.08 | 8.18 | 16.67 | 37.50 | 30.00 | 46.67 | 0.00 | 70.00 | 25.00 | 30.00 |
| | | Beam | 78.97 | 89.46 | 75.74 | 26.92 | 18.18 | 16.67 | 37.50 | 30.00 | 43.33 | 16.67 | 70.00 | 15.00 | 29.33 |
| | | Nucleus | 75.30 | 78.86 | 73.68 | 23.08 | 18.18 | 16.67 | 37.50 | 20.00 | 30.00 | 33.33 | 60.00 | 25.00 | 26.67 |
| | | Opera | 64.50 | 60.20 | 70.67 | 7.69 | 4.55 | 0.00 | 12.50 | 0.00 | 16.67 | 0.00 | 0.00 | 10.00 | 7.33 |
| | | VCD | 76.70 | 80.69 | 75.08 | 23.08 | 36.36 | 3.89 | 25.00 | 30.00 | 33.33 | 16.67 | 60.00 | 30.00 | **32.67** |
| | | Ours | **80.17** | 87.37 | **78.05** | 23.08 | 18.18 | 27.78 | 62.50 | 30.00 | 35.67 | 0.00 | 70.00 | 30.00 | 31.33 |
| InternVL | 26B | Greedy | 85.83 | 82.83 | 86.45 | 42.31 | 36.36 | 27.78 | 25.00 | 30.00 | 80.00 | 33.33 | 80.00 | 45.00 | 48.00 |
| | | Beam | 86.80 | 87.45 | 86.68 | 38.46 | 45.45 | 22.22 | 37.50 | 50.00 | 83.33 | 50.00 | 70.00 | 45.00 | 50.67 |
| | | Nucleus | 81.23 | 79.45 | 81.76 | 50.00 | 31.82 | 27.78 | 12.50 | 60.00 | 70.00 | 33.33 | 60.00 | 25.00 | 44.00 |
| | | Opera | 86.30 | 84.44 | 86.66 | 42.31 | 27.27 | 16.67 | 25.00 | 30.00 | 76.67 | 50.00 | 70.00 | 50.00 | 45.33 |
| | | VCD | 81.70 | 80.08 | 82.18 | 30.77 | 36.36 | 11.11 | 12.50 | 50.00 | 66.67 | 50.00 | 50.00 | 55.00 | 42.00 |
| | | Ours | **86.93** | 83.70 | **87.53** | 42.31 | 50.00 | 44.44 | 12.50 | 30.00 | 83.33 | 50.00 | 70.00 | 40.00 | **51.33** |

environment, holistic description, and others. GPT-4 measures the severity of hallucination in a range of 0 to 7 and the higher score denotes less hallucination.

## 4.1 Evaluation Results

**Overview**  We summarize our comprehensive experimental results across six LMMs, six decoding methods, on six benchmarks in a spider chart format for visibility of the improvements in Fig. 3. As in the figure, CODE generally shows competent performance and consistent results among different measurements and benchmarks. We delve into each result in detail in the subsections below.

**Results on Discriminative Benchmarks**  We conduct hallucination evaluation across 6 LMM baselines on two discriminative benchmarks. POPE [35] focuses on object-level hallucinations, where its prompts consist of asking about the existence of objects, such as *"Is there <something> in the image?"*. Here, we exclude trivial splits and test with the most challenging subset: *adversarial* split, which requires models to identify highly-relevant objects to correctly answer yes/no. As shown in Table 1, both the accuracy and F1 score from CODE lead to the best performance in 6 LLM baselines, among 6 decoding methods, which indicates that our decoding method can properly address object hallucination within simple binary question format.

As a more sophisticated visual assessment, we compare our method on MMVP benchmark [53]. Beyond object hallucination, the benchmark is divided into 9 systematic visual patterns within CLIP-blind pairs. These pairs consist of two images that clearly display visual differences, yet CLIP [48] struggles to discriminate between them. The test subject models only receive scores when they correctly identify both images in the pair, requiring precise visual understanding. On the right part of Table. 1, we can observe a remarkable gain in average, indicating that CODE effectively enhances visual consistency through the contrastive mechanism using self-generated descriptions.

**Results on GPT-aided Benchmarks**  Although the discriminative datasets are intuitive benchmarks for evaluating the degree of object-level hallucination, they are limited in measuring a deeper under-

Table 2: GPT-aided evaluation results among 6 LMMs and decoding methods on generative benchmarks (LLaVA-QA90 [42]: score ratio for GPT answer / MMHal-Bench [52]: score rated by GPT).

| Model | #Param | Decoding | LLaVA-QA90 | | | | MMHal-Bench | | | | | | | | | |
|---|---|---|---|---|---|---|---|---|---|---|---|---|---|---|---|---|
| | | | Conv | Detail | Comp | Overall↑ | Attr | Adv | Comp | Count | Rel | Env | Hol | Other | Overall↑ | Hal↓ |
| LLaVA-1.5 | 13B | Greedy | 79.5 | 77.1 | 90.3 | 82.4 | 3.17 | 1.92 | 2.42 | 2.25 | 1.75 | 3.50 | 1.92 | 2.17 | 2.39 | 52.08 |
| | | Beam | 81.8 | 77.7 | 90.7 | **83.5** | 2.92 | 1.67 | 2.92 | 1.83 | 1.50 | 2.92 | 2.33 | 2.58 | 2.33 | 53.13 |
| | | Nucleus | 70.4 | 78.3 | 89.9 | 79.3 | 2.08 | 0.75 | 1.83 | 2.33 | 1.92 | 3.42 | 1.67 | 2.25 | 2.03 | 60.42 |
| | | Opera | 82.4 | 73.5 | 85.4 | 80.7 | 3.08 | 0.75 | 2.58 | 1.83 | 1.92 | 3.08 | 2.08 | 2.42 | 2.22 | 55.00 |
| | | VCD | 72.1 | 76.4 | 89.5 | 79.3 | 2.83 | 1.58 | 2.33 | 2.42 | 1.67 | 2.67 | 1.83 | 2.92 | 2.28 | 54.00 |
| | | Ours | 81.8 | 77.7 | 90.7 | **83.5** | 3.00 | 1.75 | 2.25 | 2.17 | 1.58 | 3.00 | 3.25 | 2.92 | **2.49** | 51.00 |
| Emu2-Chat | 14B | Greedy | 81.1 | 59.0 | 88.1 | 76.4 | 3.17 | 1.08 | 2.50 | 2.58 | 2.17 | 2.83 | 2.83 | 2.83 | 2.50 | 39.58 |
| | | Beam | 81.7 | 54.0 | 87.9 | 74.8 | 3.00 | 0.58 | 1.83 | 2.75 | 2.58 | 3.33 | 3.17 | 2.42 | 2.46 | 39.58 |
| | | Nucleus | 78.6 | 57.9 | 87.3 | 74.9 | 2.50 | 0.67 | 2.83 | 2.83 | 1.92 | 3.25 | 3.00 | 3.17 | 2.52 | 37.50 |
| | | Opera | 82.6 | 54.5 | 66.2 | 68.0 | 2.75 | 1.08 | 1.75 | 2.75 | 2.08 | 2.67 | 2.92 | 2.58 | 2.33 | 43.75 |
| | | VCD | 80.7 | 56.8 | 87.6 | 75.5 | 2.92 | 0.75 | 1.83 | 3.00 | 2.08 | 2.92 | 2.92 | 2.92 | 2.42 | 42.00 |
| | | Ours | 82.1 | 59.6 | 89.5 | **77.3** | 2.92 | 1.08 | 2.42 | 2.83 | 2.25 | 3.33 | 3.00 | 2.92 | **2.59** | 37.50 |
| IXC2-VL | 7B | Greedy | 83.0 | 81.3 | 90.3 | 84.9 | 3.58 | 3.08 | 3.17 | 2.17 | 4.08 | 3.50 | 2.75 | 3.08 | 3.17 | 29.17 |
| | | Beam | 79.1 | 80.2 | 90.4 | 83.1 | 3.50 | 3.25 | 2.83 | 2.25 | 4.33 | 3.83 | 3.08 | 3.08 | 3.23 | 28.13 |
| | | Nucleus | 86.5 | 76.5 | 88.3 | 84.1 | 3.25 | 2.75 | 2.92 | 2.92 | 3.25 | 3.42 | 2.92 | 3.00 | 3.05 | 33.00 |
| | | Opera | 78.8 | 79.5 | 86.9 | 81.7 | 3.67 | 3.25 | 2.92 | 2.42 | 4.25 | 4.17 | 3.00 | 3.00 | 3.33 | 27.08 |
| | | VCD | 79.5 | 82.1 | 92.0 | 84.5 | 3.58 | 2.42 | 3.08 | 2.50 | 3.33 | 3.92 | 2.33 | 2.75 | 2.99 | 32.00 |
| | | Ours | 85.3 | 79.4 | 92.9 | **86.0** | 3.67 | 3.25 | 3.17 | 3.00 | 3.67 | 3.92 | 3.33 | 3.67 | **3.46** | 25.00 |
| LLaVA-NeXT | 34B | Greedy | 80.0 | 95.7 | 97.9 | 90.7 | 2.75 | 3.92 | 3.50 | 2.75 | 2.83 | 3.08 | 3.50 | | 3.30 | 34.00 |
| | | Beam | 87.7 | 97.2 | 99.2 | 94.5 | 3.33 | 3.83 | 2.83 | 2.58 | 3.17 | 3.92 | 2.67 | 3.75 | 3.26 | 35.42 |
| | | Nucleus | 83.2 | 88.7 | 98.3 | 90.0 | 2.50 | 3.92 | 3.50 | 1.33 | 3.92 | 4.42 | 2.33 | 2.75 | 3.08 | 40.63 |
| | | Opera | - | - | - | - | - | - | - | - | - | - | - | - | - | - |
| | | VCD | 86.4 | 93.8 | 96.6 | 92.1 | 3.25 | 3.75 | 3.08 | 3.08 | 2.17 | 3.92 | 2.50 | 3.58 | 3.16 | 39.58 |
| | | Ours | 89.2 | 99.1 | 98.3 | **95.3** | 2.92 | 4.08 | 3.92 | 2.17 | 2.92 | 4.67 | 2.92 | 3.83 | **3.43** | 34.00 |
| Yi-VL | 34B | Greedy | 80.2 | 76.4 | 88.9 | 81.9 | 2.67 | 0.00 | 3.08 | 1.00 | 2.00 | 2.25 | 1.83 | 1.17 | 1.75 | 65.63 |
| | | Beam | 70.7 | 75.7 | 85.9 | 77.5 | 2.92 | 0.42 | 3.33 | 0.92 | 2.25 | 2.33 | 1.83 | 1.58 | **1.95** | 60.42 |
| | | Nucleus | 76.2 | 81.2 | 82.7 | 79.9 | 2.33 | 1.42 | 2.58 | 1.08 | 2.50 | 2.42 | 1.33 | 1.08 | 1.84 | 62.50 |
| | | Opera | 63.6 | 29.1 | 16.0 | 36.3 | 2.50 | 1.25 | 1.83 | 0.83 | 1.00 | 1.83 | 1.92 | 1.50 | 1.58 | 61.50 |
| | | VCD | 66.9 | 76.7 | 87.3 | 76.7 | 2.58 | 0.33 | 3.00 | 1.58 | 2.42 | 2.00 | 1.92 | 0.92 | 1.84 | 62.50 |
| | | Ours | 73.9 | 83.6 | 89.4 | **82.1** | 2.75 | 0.58 | 3.00 | 1.00 | 2.50 | 2.50 | 1.33 | 1.00 | 1.83 | 65.63 |
| InternVL | 26B | Greedy | 79.2 | 93.2 | 89.1 | 86.6 | 3.92 | 2.58 | 2.67 | 2.42 | 3.17 | 3.75 | 3.25 | 3.42 | 3.15 | 33.33 |
| | | Beam | 83.7 | 93.3 | 91.6 | 89.3 | 3.83 | 2.67 | 3.83 | 2.50 | 4.00 | 3.75 | 2.75 | 3.58 | 3.36 | 31.25 |
| | | Nucleus | 75.1 | 95.1 | 91.1 | 86.4 | 2.92 | 3.25 | 2.75 | 3.50 | 2.75 | 3.33 | 3.33 | 3.25 | 3.14 | 37.50 |
| | | Opera | 81.0 | 94.6 | 91.9 | 88.7 | 3.58 | 2.83 | 3.00 | 2.92 | 4.17 | 3.83 | 2.92 | 3.33 | 3.32 | 32.29 |
| | | VCD | 83.9 | 92.6 | 89.5 | 88.3 | 3.67 | 2.17 | 3.50 | 2.50 | 2.67 | 3.92 | 2.08 | 3.00 | 2.94 | 42.00 |
| | | Ours | 83.1 | 103.3 | 91.9 | **92.2** | 4.25 | 2.92 | 3.5 | 2.92 | 4.17 | 3.83 | 2.92 | 3.67 | **3.52** | 30.21 |

standing of whether the given LMM responses encompass contextual or sequential hallucination. To probe the expanded effectiveness of our method beyond simple multiple choice tests, we utilize two generative benchmarks that can verify model responses at the sentence level. As in Table. 2, CODE generally outperforms the overall score of LLaVA-QA90 [42], showing up to $+13.7\%$ improvements than other CD methods. Additionally, we compare our models in MMHal-Bench [52] specialized to evaluate hallucination effects sourced from more challenging image-question pairs— OpenImages [50]. As in the result, our method generally not only improves overall average score with consistent results among 6 other baseline LMMs, but also effectively mitigates the hallucination ratio. Through combinatorial results from both discriminative and generative benchmarks, we corroborate the robustness and efficacy of our proposed method in addressing both object-level and contextual hallucinations, thereby ensuring more reliable LMM responses.

## 4.2 Analyses on CODE

**Ablation Study** Previous CD-based methods [34, 30, 12] require heuristic choices to control the degree of amplification for logit variation $\alpha$ and penalizing parameter $\beta$ that filter out the implausible next-tokens in adaptive plausibility constraint. To tackle it, we proposed two regulation methods that can dynamically control information flow in CODE and token candidate pool, respectively: (i) dynamic restriction (DR), $\alpha_t$ in Eqn. 2 and (ii) adaptive information constraint (AIC), $\beta_t$ in Eqn. 4. To validate the effectiveness of such adaptive regulations built in CODE, we conducted ablation study on them.

Table 3: Ablation study on $\alpha_t$ (DR) and $\beta_t$ (AIC) for (abbreviated) LV1.5 [40], LV-N [41], IVL1.5 [8] on two benchmarks [53, 42]. We report overall scores for the benchmarks. ✗ indicates fixed hyper-parameter for $\alpha$ and $\beta$.

| DR | AIC | MMVP | | | LLaVA-Bench | | |
|---|---|---|---|---|---|---|---|
| | | LV1.5 | LV-N | IVL1.5 | LV1.5 | LV-N | IVL1.5 |
| ✗ | ✗ | 28.67 | 32.67 | 45.33 | 66.4 | 76.0 | 81.0 |
| ✗ | ✓ | 32.00 | 41.33 | 48.00 | 67.9 | 81.6 | 83.5 |
| ✓ | ✗ | 29.33 | 37.33 | 48.67 | 69.0 | 79.7 | 82.4 |
| ✓ | ✓ | **34.00** | **42.67** | **51.33** | **72.6** | **85.1** | **85.7** |

We implement baselines with same CODE framework using self-generated descriptions as visual counterparts, but with default settings of $\alpha{=}1.0$ and $\beta{=}0.1$. As in Table. 3, either use of DR or AIC can enhance the benchmarks than the fixed $\alpha$ and $\beta$. Our CODE implementation that utilizes both DR and AIC to dynamically restrict information flow exhibits the best results among the baselines.

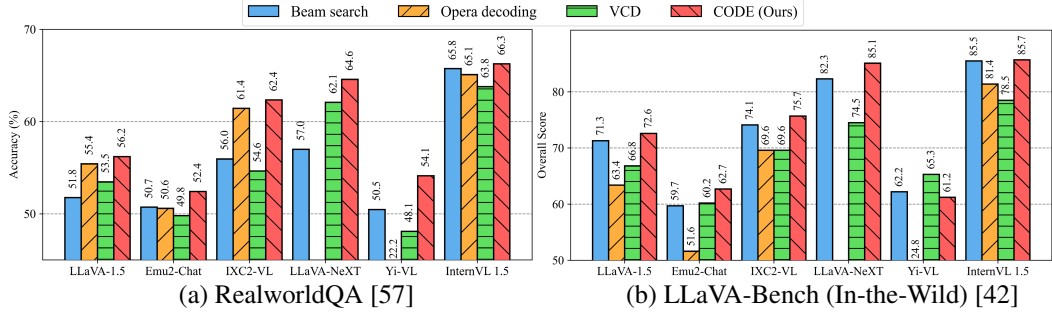

**Question:** What is the brand of the yogurt flavored with blueberry?

**GT Answer:** The brand of the blueberry-flavored yogurt is Fage.

| | | | | | | | | | | | |
|---|---|---|---|---|---|---|---|---|---|---|---|
| Original Tokens | The | brand | in | the | image | is | from | the | brand | " | Yoplait … |
| CODE Tokens | The | yogurt | in | the | image | is | from | the | brand | " | Fage … |
| Visual content $logit_v$ | 20.81 | 21.34 | 20.83 | 27.03 | 22.47 | 22.84 | 18.42 | 19.78 | 20.97 | 18.63 | **15.34** … |
| | 20.81 | 21.13 | 20.83 | 27.03 | 22.47 | 22.84 | 18.42 | 19.78 | 20.97 | 18.63 | 15.02 … |
| Description-only $logit_d$ | 16.98 | 20.14 | 18.69 | 25.55 | 21.00 | 23.00 | 16.41 | 19.23 | 19.59 | 17.53 | **15.27** … |
| | 16.98 | 15.82 | 18.69 | 25.55 | 21.00 | 23.00 | 16.41 | 19.23 | 19.59 | 17.53 | 13.02 … |
| CODE $logit_{code}$ | 22.01 | 21.85 | 22.17 | 28.34 | 23.40 | 22.72 | 19.78 | 20.26 | 22.06 | 19.42 | 15.39 … |
| | 22.01 | 23.38 | 22.17 | 28.34 | 23.40 | 22.72 | 19.78 | 20.26 | 22.06 | 19.42 | **16.66** … |

Figure 4: An example of token-level case study for CODE. Each row indicates the logit score from visual content $logit_v$, comprehensive description $logit_d$, CODE applied $logit_{code}$, respectively.

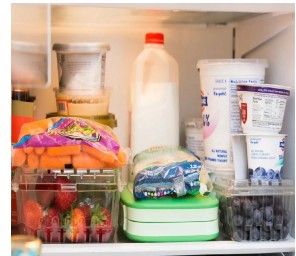

(a) RealworldQA [57]

(b) LLaVA-Bench (In-the-Wild) [42]

Figure 5: Additional experiments on In-the-Wild benchmarks. Note that, unlike other datasets, OPERA [23] fails to generate consistent responses in real-world datasets using Yi-VL [62].

**Computational Analysis** We utilize comprehensive descriptions from models as additional information for contrasting with visual contents, thereby leading to computational loads similar to other decoding methods. To analyze the computation, we compare the token throughput (token/s) and decoding latency (ms/token) with other CD-based methods on 8 NVIDIA RTX A6000 GPUs as in Table. 4.

Table 4: Computational analysis on decoding throughput and latency among CD-based methods. We compare three different model sizes.

| | Throughput (token/s)↑ | | | Latency (ms/token)↓ | | |
|---|---|---|---|---|---|---|
| | VCD | OPERA | CODE | VCD | OPERA | CODE |
| 7B [16] | 5.62 | 1.23 | 3.66 | 177.99 | 809.73 | 272.92 |
| 14B [51] | 4.04 | 1.04 | 2.82 | 247.6 | 960.14 | 354.09 |
| 34B [41] | 3.61 | oom | 2.81 | 277.27 | oom | 355.81 |

**Token-level Case Study** As illustrated in Fig. 4, to verify whether the proposed CODE effectively mitigates object hallucination, we analyze the output logit values of LMM [41] at the token-level case study, with a greedy search as the baseline. The first two rows in the table indicate the original greedy decoded tokens which are elected based on high $logit_v$ from the visual content and CODE output tokens, respectively. As in the figure, we can observe that visual hallucination occurs at the *"Yoplait"* token highlighted in red. For relatively easy tokens at the beginning of sentence, $logit_{code}$ produces identical decisions maintaining consistency with $logit_v$, which indicates the amplification of logit variation is effectively adjusted due to similar prediction distributions from visual contents and description-only information. However, at the hallucination-occurred time step, logit scores are deviated between the two information, resulting in a more confusing state to identify between GT token *"Fage"* and hallucinatory *"Yoplait"*. In our framework, *"Fage"* is relatively more amplified from 15.02 to 16.66 than *"Yoplait"*, which changes from 15.34 to 15.30. By simultaneously considering both token-level and distributional prediction over the vocabulary, CODE changes the wrong next-token output to correct one, mitigating hallucination. For more case studies, please refer Appendix. D.

**Additional Experiments on In-the-Wild** Contemporary open-sourced LMMs are fine-tuned with various combinations of vision-language datasets [7, 28, 25], mostly composed of COCO-sourced visual images [38] and their curated instruction. Although the existing hallucination benchmarks intentionally convert question queries to assess model robustness against inconsistency, the visual contents in benchmarks are limited to in-distribution COCO images. To validate our method in more challenging and real-world scenarios, we compared baselines on LLaVA-Bench (In-the-wild) [42] and RealworldQA [57] as in Fig. 5 and achieved competent performance (case studies in Appendix. F).

# 5 Discussion and Limitation

Albeit the computational analysis in Table. 4, as one of limitations, our contrastive decoding method requires additional computational resources than the use of vanilla decoding. However, considering an essential ongoing research topics and developments [60, 15] aimed at mitigating the negative effects of hallucination problems in both LLMs and LMMs, our work contributes important societal impacts towards more real-world applicability and robust AI system.

# 6 Conclusion

We present COuntering DEscription Contrastive Decoding (CODE), a novel and training-free decoding method to mitigate hallucination in Large Multi-modal Models. By utilizing self-generated descriptions as corrective references during the decoding phase, CODE dynamically adjusts the information flow for next-token predictions, enhancing the coherence and informativeness of responses while reducing the cross-modal inconsistency. Extensive experiments demonstrate that CODE effectively decreases hallucinations across various benchmarks and contemporary LMMs, significantly improving contextual relevance and response alignment with visual contents.

## Acknowledgments

This work was partially supported by two funds: IITP grant funded by the Korea government (MSIT) (RS-2022-II220984) and Center for Applied Research in Artificial Intelligence (CARAI) grant funded by DAPA and ADD (UD230017TD).

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

# A  Instruction for Comprehensive Description

In generating a comprehensive description for the given visual content, we aim to obtain as detailed a description as possible, ensuring that the response fully spans the visual representation space, even though it may not be entirely feasible as discussed in 3.1. The specific prompt instruction used to describe the image contents in detail is described in Table. 5.

---

**Comprehensive Description Prompt:**
Provide a detailed description of the image, covering all visible elements and their interactions, so as to thoroughly answer any potential questions about the image.

---

Table 5: A simple instruction prompt for generating comprehensive description.

# B  Detailed Algorithm for CODE

We describe the complete details of CODE implementation for better understanding in Algorithm. 1.

---

**Algorithm 1** COuntering DEscription Contrastive Decoding

---

**Require:** LVLM $M_\theta$, Visual Content $v$, Text Query $x$, Comprehensive Description Prompt $x_0$, Target Token Length $T$

1: $d \leftarrow M_\theta(v, x_0)$    ▷ Step 1: Generate comprehensive description $d$ for the given visual content
2: Initialize $t \leftarrow 1$
3: **while** $t < T$ **do**    ▷ Step2: CODE decoding
4:     $p_v = \text{Softmax}\left[\text{logit}_\theta(y_t \mid v, x, y_{<t})\right]$    ▷ Compute $p_v$
5:     $p_d = \text{Softmax}\left[\text{logit}_\theta(y_t \mid d, x, y_{<t})\right]$    ▷ Compute $p_d$
6:     $\mathcal{D}_{\text{bd}}(p_v \| p_d) = \frac{1}{2} \sum_{i=1}^{n} (p_{v,i} + p_{d,i}) \log_2(|p_{v,i} - p_{d,i}|^k + 1)$    ▷ Bounded Divergence $\mathcal{D}_{\text{bd}}$
7:     Set $\alpha_t \leftarrow 1 - \mathcal{D}_{\text{bd}}(p_{v,t} \| p_{d,t})$    ▷ Set $\alpha_t$ for Dynamic Restriction
8:     Set $\beta_t \leftarrow \mathcal{D}_{\text{bd}}(p_{v,t} \| p_{d,t})$    ▷ Set $\beta_t$ for Adaptive Information Constraint
9:     $\mathcal{V}_{\text{head}}(y_{<t}) = \{y_t \in \mathcal{V} : p_\theta(y_t \mid v, x, y_{<t}) \geq \beta_t \max_w p_\theta(w \mid v, x, y_{<t})\}$    ▷ Set $\mathcal{V}_{\text{head}}$
10:     $p_{\text{code}} = \text{Softmax}[(1 + \alpha_t)\text{logit}_\theta(y_t \mid v, x, y_{<t}) - \alpha_t\text{logit}_\theta(y_t \mid d, x, y_{<t})]$
11:     $p_{\text{code}}(y_t \mid y_{<t}) = 0, \text{if } y_t \notin \mathcal{V}_{\text{head}}$    ▷ Apply Adaptive Information Constraint
12:     $y_t = \text{argmax}(p_{\text{code}})$    ▷ Select Token
13:     Set $t \leftarrow t + 1$
14: **end while**

---

# C  Benchmark Details

**Discriminative Benchmarks**

- **POPE** [35] is a widely used benchmark designed for evaluating object-level hallucination, which can be split into the three subset categories based on how to select object replacements: (i) random, randomly sampled objects (ii) popular, top-$k$ frequent objects not existing in the image, and (iii) adversarial, top-$k$ objects that have high co-occurrence. The number of images is 500 and each image has 6 questions along with the subsets, making a total of 9000 images. In this work, we only consider adversarial split, which is the most challenging subset in POPE benchmark.

- **MMVP** [53] includes 300 images with 9 different visual patterns that CLIP model [48] struggles to identify the visual differences (CLIP-paired images): Orientation and Direction (⊘), Presence of Specific Features (🔍), State and Condition (🔄), Quantity and Count (↕️), Positional and Relational Context (📍), Color and Appearance (🎨), Structural and Physical Characteristics (⚙️), Text (A), Viewpoint and Perspective (📷). It follows multiple selection tests, but uses GPT-4 to map the model response to the answer options.

- **RealworldQA** [57] is recently introduced benchmarks for evaluating basic real-world understanding for multi-modal models. It consists of total 765 anonymized outdoor (mostly taken from vehicles) and indoor images with multiple selection questions.

**Generative Benchmarks**

- **LLaVA-QA90 & LLaVA-Bench (In-the-Wild)** [42] consist of three subset response types for each image: (i) Conversation, which is conversation format between the user and assistant answering the vision-related questions for the given images, (ii) Detailed description, which requires detailed description for the given image scene, and (iii) Complex reasoning, which involves in-depth reasoning questions for the image. The former benchmarks sourced from COCO images (total 30 images with 90 questions), while the latter benchmarks are gathered from web for challenging domain situations (total 24 images with 60 questions).

- **MMHal-Bench** [52] is specially focused on penalizing hallucinations. It has total 96 image-question pairs composed with 8 question categories for 12 objects: Object attribute (Attr), Adversarial object (Adv), Comparison (Comp), Counting (Count), Spatial relation (Rel), Environment (Env), Holistic description (Hol), and Others (Other). As like in the above LLaVA-Bench benchmarks, MMHal-Bench also utilize GPT-4 to analyze and rate the model responses and score in a range of 0 to 7.

## D   Additional Token-level Case Study

As additional token-level case studies, we explore how the logit information changes during our CODE decoding phase in other examples. As a first example, LLaVA-NeXT [41] struggles to distinguish between Haleakala National Park and Diamond Head, both located in Hawaii, and predicts the former during inference with the vanilla decoding method. Using our CODE decoding method, as shown in Fig. 6, the information flow of *"Haleakala"* token is curbed, inducing a token inversion to *"Diamond"*, which matches the ground truth word. This occurs because the logit variation is dynamically adjusted based on both token-level and distributional information.

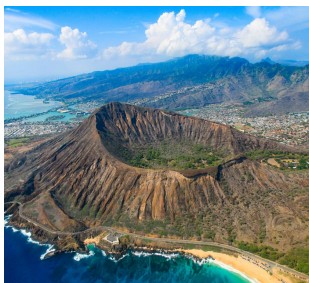

**Question:** What is the name of this famous sight in the photo?

**GT Answer:** The famous sight in the photo is Diamond Head.

| Original Tokens | The | famous | sight | in | the | photo | is | Haleakala | ... |
|---|---|---|---|---|---|---|---|---|---|
| CODE Tokens | The | famous | sight | in | the | photo | is | Diamond | ... |
| Visual content $\text{logit}_v$ | 23.42 | 20.89 | 24.78 | 26.45 | 30.55 | 28.16 | 27.05 | **18.70** | ... |
| | 23.42 | 20.89 | 24.78 | 26.45 | 30.55 | 28.16 | 27.05 | 17.45 | ... |
| Description-only $\text{logit}_d$ | 18.89 | 19.06 | 23.81 | 24.86 | 28.64 | 26.34 | 24.23 | **17.95** | ... |
| | 18.89 | 19.06 | 23.81 | 24.86 | 28.64 | 26.34 | 24.23 | 11.86 | ... |
| CODE $\text{logit}_{code}$ | 25.57 | 21.91 | 25.66 | 27.89 | 32.29 | 29.95 | 28.58 | 19.32 | ... |
| | 25.57 | 21.91 | 25.66 | 27.89 | 32.29 | 29.95 | 28.58 | **22.06** | ... |

Figure 6: Additional token-level case study for analysis.

In another example illustrated in Fig. 7, we show how the adaptive information constraint prevent from rewarding implausible tokens, thus suppress hallucinatory prediction during contrastive decoding. The original tokens from $\text{logit}_v$ predict the correct answer at the hallucinatory time step, highlighted in bold. In this case, the original prediction should be preserved, and token inversion should not occur. By our CODE decoding method, $\beta_t$ dynamically controls the adaptive information constraint, so that those hallucination tokens'(*i.e., "four"* and *"dragon"*) logit values are cut off to $-\infty$ and removed from candidate token pool.

## E   Further Discussion on Broader Impact

We proposed CODE, which can be seamlessly integrated into LMMs without additional training. There is still a lot of room for progress and mitigation of hallucination issues, as our method cannot assure 100% removal of hallucinations. However, by providing more coherent and contextually accurate responses, our work can potentially be integrated into real-world applications, making user interactions with AI in customer service, education, and personal assistance more effective and satisfying in the near future.

Furthermore, enhanced accuracy and reliability using our method can reduce hallucinations in LMMs, improving the accuracy of AI-generated descriptions in critical fields such as autonomous

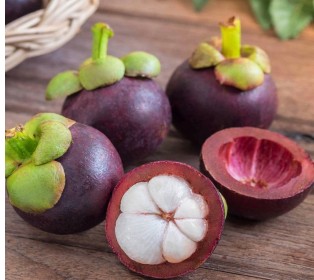

**Question:** How many uncut fruits are in the image?
**GT Answer:** There are three uncut fruits in the image.

| $\beta_t$ | 0.27 | 0.01 | **0.80** | 0.11 | 0.00 | **0.80** | 0.05 | 0.00 | 0.01 | 0.18 |
|---|---|---|---|---|---|---|---|---|---|---|
| Original Tokens | There | are | three | unc | ut | fruits | in | the | image | . |
| Description Tokens | There | are | four | unc | ut | dragon | in | the | image | . |
| Visual content $\text{logit}_v$ | 22.56 | 26.53 | 23.09 | 22.16 | 29.47 | 21.17 | 25.41 | 30.00 | 27.42 | 25.50 |
| | 22.56 | 26.53 | 21.66 | 22.16 | 29.47 | 14.63 | 25.41 | 30.00 | 27.42 | 25.50 |
| Description-only $\text{logit}_d$ | 20.47 | 27.09 | 21.59 | 23.95 | 28.20 | 20.75 | 25.34 | 28.36 | 26.50 | 21.59 |
| | 20.47 | 27.09 | 24.28 | 23.95 | 28.20 | 21.80 | 25.34 | 28.36 | 26.50 | 21.59 |
| CODE Tokens | There | are | three | unc | ut | fruits | in | the | image | . |
| CODE $\text{logit}_{code}$ | 24.10 | 25.98 | 23.39 | 20.56 | 30.73 | 21.25 | 25.47 | 31.63 | 28.34 | 28.69 |
| | 24.10 | 25.98 | -inf | 20.56 | 30.73 | -inf | 25.47 | 31.63 | 28.34 | 28.69 |

Figure 7: Additional token-level case study for analysis.

driving, robotics, healthcare, and augmented reality. This advancement not only enhances practical applications but also significantly benefits the research community working on hallucination, an area that is not yet fully explored. Our contributions can help pave the way for deeper understanding and new research directions to address these challenges, for more trustworthy AI systems.

# F Additional Qualitative Results

Discriminative Capability Case Study: MMVP.

**Image:**

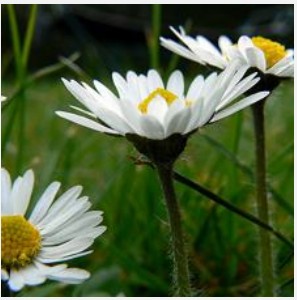

**Question:**
Is the following statement correct: In the image, there is one daisy that is under the shadow of a taller daisy.
(a) Correct (b) Incorrect

**Ground-truth:**
(b)

**InternLM-XComposer2 (Opera):**
The answer is (a).

**InternLM-XComposer2 (VCD):**
The answer is (a).

**InternLM-XComposer2 (Ours):**
The answer is (b).

**Image:**

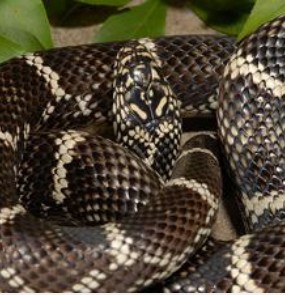

**Question:**
In the image, is the snake's head oriented facing upwards or directed forwards?
(a) Upwards (b) Forward

**Ground-truth:**
(a)

**InternVL 1.5 (Opera):**
The snake's head is oriented (b) forward.

**InternVL 1.5 (VCD):**
The snake's head is directed forwards, so the correct answer is (b) Forward.

**InternVL 1.5 (Ours):**
The snake's head is oriented facing upwards, so the correct answer is (a) Upwards.

**Image:**

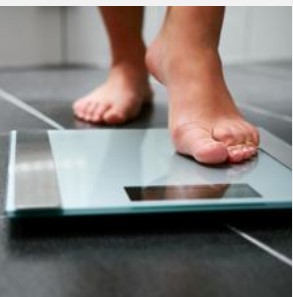

**Question:**

Is the person in the picture already standing on the weighing scale?
(a) Yes (b) No

**Ground-truth:**

(b)

**InternVL 1.5 (Opera):**

(a) Yes

**InternVL 1.5 (VCD):**

Yes, the person in the picture is already standing on the weighing scale.

**InternVL 1.5 (Ours):**

The person in the picture is not yet standing on the weighing scale. The answer is (b) No.

---

**Image:**

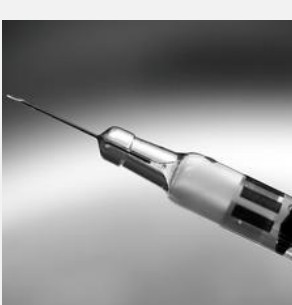

**Question:**

Is the needle pointing up or down?
(a) Up (b) Down

**Ground-truth:**

(a)

**LLaVA-NeXT (VCD):**

The needle of the syringe is pointing down.

**LLaVA-NeXT (Ours):**

The needle in the image is pointing up.

---

**Image:**

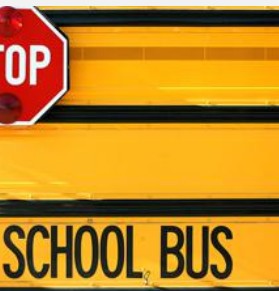

**Question:**

Based on the image, is the following statement correct: We cannot see the window on the school bus?
(a) Correct (b) Incorrect

**Ground-truth:**

(a)

**LLaVA-NeXT (VCD):**

(b) Incorrect. The window of the school bus is visible in the image, located just above the "School Bus" sign.

**LLaVA-NeXT (Ours):**

The statement "We cannot see the window on the school bus" is (a) Correct. In the image provided, there is no visible window on the school bus. The view is focused on the back of the bus where the stop sign and the words "SCHOOL BUS" are visible, but the window is not shown.

Table 6: More case studies within MMVP, using InternLM-XComposer2, InternVL 1.5 and LLaVA-NeXT.

Discriminative Capability Case Study: RealworldQA.

**Image:**

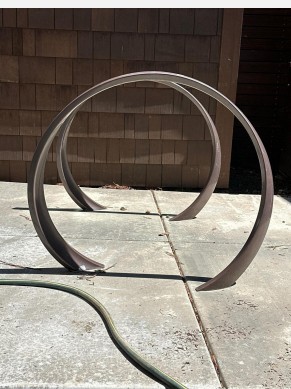

**Question:**
How many circular objects are in the image?
A. There are 2 circular objects.
B. There is 1 circular object.
C. There are 3 circular objects.
Please answer directly with only the letter of the correct option and nothing else.

**Ground-truth:**
A

**InternLM-XComposer2 (Opera):**
The answer is C.

**InternLM-XComposer2 (VCD):**
The answer is C.

**InternLM-XComposer2 (Ours):**
A.

**Image:**

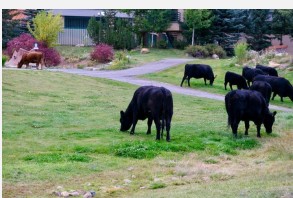

**Question:**
How many animals are in this picture?
A. There are eight animals in this picture.
B. There are nine animals in this picture.
C. There is one animal in this picture.
Please answer directly with only the letter of the correct option and nothing else.

**Ground-truth:**
B

**InternVL 1.5 (Opera):**
A

**InternVL 1.5 (VCD):**
A

**InternVL 1.5 (Ours):**
B

**Image:**

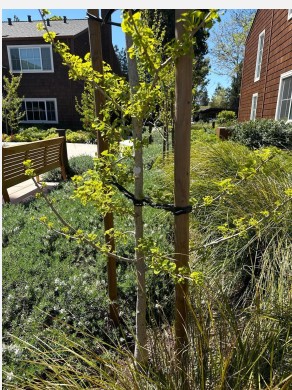

**Question:**
Where is the black material relative to the tree?
A. The black material is at the top of the tree.
B. The black material is at the midpoint or halfway on the tree.
C. There is no black material.
Please answer directly with only the letter of the correct option and nothing else.

**Ground-truth:**
B

**LLaVA-NeXT (VCD):**
A

**LLaVA-NeXT (Ours):**
B

Table 7: More case studies within RealworldQA, using InternLM-XComposer2, InternVL 1.5 and LLaVA-NeXT.

Generative Capability Case Study: LLaVA-Bench (In-the-Wild).

**Question:** Describe this photo in detail.

**Image:**

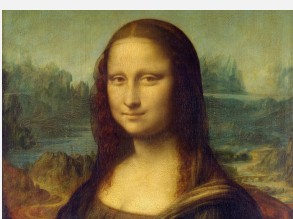

**InternLM-XComposer2 (Opera):**

The painting you're referring to is the famous "Mona Lisa" by Leonardo da Vinci. It is an oil painting on a poplar wood panel. The artwork depicts a woman, believed to be Lisa Gherardini, the wife of a Florentine merchant named Francesco del Giocondo. The woman is portrayed in a three-quarter view, with her left hand resting on her lap and her right hand holding a veil. She is wearing a dress with a high collar and a pearl necklace. The background features a landscape with mountains and a body of water. The painting is known for its enigmatic smile and the use of sfumato, a painting technique that creates a soft, hazy effect. The "Mona Lisa" is considered a masterpiece of the Italian Renaissance and is housed in the Louvre Museum in Paris, France.

**InternLM-XComposer2 (VCD):**

The image features a painting of the renowned Mona Lisa, an oil on wood artwork. The central figure is a woman with long red hair and a headband, wearing a blue dress adorned with gold accents. Her face is captured in close-up detail, showcasing her distinct smile and enigmatic gaze. The background is a landscape painted in earth tones, adding depth to the composition. This painting is widely recognized as a masterpiece, capturing the essence of the Renaissance era with its realistic portrayal and intricate details.

**InternLM-XComposer2 (Ours):**

The painting you're referring to is the famous "Mona Lisa" by Leonardo da Vinci. It is an oil painting on a wooden panel, measuring approximately 77 x 53 centimeters. The painting is a portrait of a woman, who is the focal point of the artwork. She is depicted wearing a dress and a headscarf, with her hair styled in a manner typical of the Italian Renaissance period. The background of the painting features a landscape with mountains and a body of water, adding depth and context to the scene. The painting is known for its realistic portrayal of the woman's features, her enigmatic smile, and the subtle interplay of light and shadow. The "Mona Lisa" is considered a masterpiece of the High Renaissance period and is housed in the Louvre Museum in Paris, France.

**Question:** Describe this photo in detail.

**Image:**

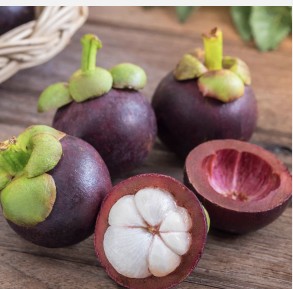

**InternVL 1.5 (Opera):**

The image shows a group of star fruit, also known as carambola or tamarind, arranged and presented. These are a kind a of sweet, fizzi, and tart tasting, and are a popular fruit. There appears a few different ways they're displayed, some whole or unpeel, some are half-cut open. There is a clear visual focus and contrast, as well as a natural and organic feel to the image.

**InternVL 1.5 (VCD):**

This image depicts several purple-colored fruits, which appear to be durian. Durian is a tropical fruit that is known for its strong, pungent aroma and creamy, sweet flesh. In the center of the image, one of the durians is cut in half, revealing the fruit's inner flesh, which is a creamy white with a star-like pattern of seeds. The cut side of the durian shows the texture of the flesh and the arrangement of the seeds within.
The fruit has a distinctive appearance with a thick, spiky green outer shell that protects the flesh inside. The surface of the fruit is textured and somewhat rough, with the color transitioning from deep purple to green at the points where the spiky projections emerge. The durians are resting on a wooden surface, which provides a rustic backdrop that contrasts with the vibrant colors of the fruit.
In the background, there is a wicker basket containing more of these fruits, suggesting that they have been gathered for consumption or sale. The basket is made of natural materials and complements the overall natural, organic feel of the image. The lighting in the photo is soft and natural, which highlights the textures and colors of the durians and adds to the overall appeal of the image.
Overall, this image presents the durian fruit in a way that showcases both its visual and textural qualities, as well as its role in a natural, possibly agricultural, setting.

**InternVL 1.5 (Ours):**

The image shows several whole and cut open mangosteens on a wooden surface. Mangosteens are tropical fruits known for their distinctive appearance and sweet, slightly acidic flavor. The fruit has a dark purple, almost black, outer skin that contrasts with the green, bulbous protrusions at the top, which are the remnants of the fruit's calyx. One of the mangosteens is cut in half, revealing the white, segmented flesh inside, which is the edible part of the fruit. The segments are arranged in a flower-like pattern around a central seed pod. The cut-open fruit provides a clear view of the fruit's structure and the contrast between the outer skin and the inner flesh. The wooden surface and the natural lighting give the image a rustic and organic feel, emphasizing the freshness of the fruit.

**Question:** Describe this photo in detail.

**Image:**

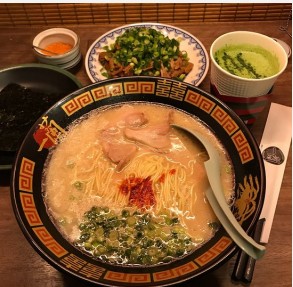

**LLaVA-NeXT (VCD):**

In the center of the image, a black bowl with vibrant red and yellow trim commands attention. It cradles a hearty serving of noodles, their creamy yellow hue contrasting with the bowl's dark color. A few pieces of meat, their rich brown tones adding to the variety of colors, are scattered throughout the bowl. Green peas peek out from under the noodles, adding a touch of freshness to the dish. To the left of the bowl, a small white cup filled with a green sauce waits patiently. Its color is a soothing contrast to the bold colors of the bowl and its contents. In the top right corner of the image, a white plate holds a small pile of chopped vegetables. Their bright green color stands out against the plate, drawing the eye to the top right corner of the image.

The arrangement of these objects creates a visually pleasing scene. The bowl of noodles is centrally placed, making it the focal point of the image. The green sauce in the white cup and the plate of chopped vegetables are positioned on either side, framing the bowl and creating a balanced composition. The colors, objects, and their relative positions all contribute to a harmonious and appetizing scene.

**LLaVA-NeXT (Ours):**

In the heart of the image, a black bowl with a vibrant red and yellow border takes center stage. It cradles a hearty serving of udon noodles, their light brown hue contrasting beautifully with the bowl's dark exterior. The noodles are not alone; they are accompanied by a generous portion of green onions, adding a pop of color and hinting at the freshness of the dish.

To the right of the bowl, a white cup filled with a green sauce waits patiently, ready to enhance the flavors of the udon noodles. The cup's pristine white color provides a stark contrast to the green sauce within, making it an eye-catching element in the composition.

In the background, a blue and white plate holds a serving of meat and green onions. The plate's colors echo those of the bowl, creating a harmonious color palette across the image.

Scattered around these main elements are various condiments and utensils, each playing their part in this culinary tableau. Their presence adds depth to the image, suggesting a meal ready to be enjoyed.

Overall, this image captures a moment of anticipation, just before the enjoyment of a delicious meal begins.

Table 8: More case studies within LLaVA-Bench (In-the-Wild), using InternLM-XComposer2, InternVL 1.5 and LLaVA-NeXT. We marked the expressions where hallucinations occurred in red.

# G Failure Cases

As discussed in Appendix. E, even if CODE shows competent performance along various benchmarks and LMM baselines. The hallucination cannot be $100\%$ eliminated. In this section, we attached some of failure cases to shed some light for future work direction. As illustrated in Table. 8, the baseline models fail to correct hallucination even with our CODE method. Upon closer examination of the model responses, it is evident that the hallucinatory responses tend to be biased towards language priors such as *"strawberry-flavored"* or *"holding a glass of beer"*. Our approach mainly uses self-generated description as contrasting reference (*i.e.,* close to the concept of self-correction), thus the strong assumption is on that the amateur model (comprehensive description) should generate not too much deviated responses from true answers.

In the failure examples, we can infer that the reliance on self-generated descriptions may not always suffice, especially when the descriptions themselves are biased or inaccurate. This indicates a need for integrating more robust mechanisms to verify and correct these biases. Additionally, enhancing the model's understanding and processing of visual content could help mitigate such issues. Future work could explore the integration of external knowledge sources and more sophisticated bias detection techniques to further reduce hallucinations and improve the overall accuracy and reliability of LMMs.

**Image:**

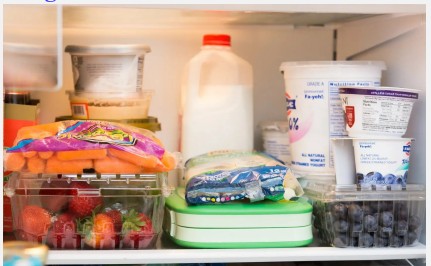

**Question:**
Is there any strawberry-flavored yogurt in the fridge?

**Generated Text:**
The image showcases the inside of a refrigerator, filled with various food items. On the top shelf, there is a carton of milk and a container of yogurt. The middle shelf contains a bag of carrots, a package of strawberries, and a blue plastic container. The bottom shelf holds a carton of blueberries and another carton of yogurt. Additionally, there are two bottles on the door of the refrigerator.

**InternLM-XComposer2 (Ours):**
Yes, there is a container of strawberry-flavored yogurt in the fridge.

**Image:**

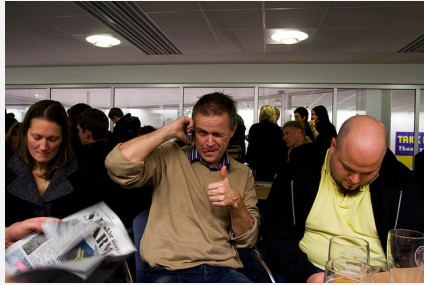

**Question:**
What is the man doing in the image?

**Generated Text:**
The image depicts a group of people gathered in a room, with some sitting and others standing. There are at least 12 people in the scene, engaged in various activities. One man is sitting at a table, holding a cell phone, while another man is sitting nearby, giving a thumbs-up gesture. There are several chairs in the room, with one near the center and another towards the right side. A dining table is also visible in the scene, with a cup placed on it. The people in the room appear to be enjoying their time together, possibly at a social event or gathering.

**LLaVA-1.5 (Ours):**
The man in the image is sitting at a table, holding a glass of beer, and giving a thumbs-up sign.

Figure 8: Failure case of InterLM-XComposer2 in LLaVA-Bench (In-the-Wild) and LLaVA-1.5 in LLaVA-QA90.

