# OpenReview forum: "CODE: Contrasting Self-generated Description to Combat Hallucination in Large Multi-modal Models"
_NeurIPS.cc/2024/Conference — NeurIPS 2024 poster_

### Official Review · Reviewer_9CKq · 2024-07-09

**Soundness:** 3
**Presentation:** 3
**Contribution:** 3
**Rating:** 5
**Confidence:** 5

**Summary:**

This paper introduce the CODE decoding method to contrast origin image with VLM-generated image description to reveal the missed or hallucinated content in naive decoding process. The method contains two innovations: 1) a Bounded Divergence guided selector to provide dynamic combining weight. 2) an adaptive information constrain based also on Bounded Divergence. The experiment results show promising and consistent performance improvement.

**Strengths:**

1. The paper is well-written and addresses a crucial problem in MLLMs.
2. The idea of contrast image and caption in VLM decoding process is novel.
3. The proposed CODE method is simple and effective on multiple benchmarks.

**Weaknesses:**

1. The dynamic information flow control does not have that much technical novelty, as it is quite close to what IBD does.
2. Like CODE, VCD also contrast image with modified inputs (i.e. corrupted images), but not discussed in detail in this paper. What makes contrasting image descriptions better than contrasting with corrupted images?
3. What makes a good image description for CODE to use is not discussed.

**Questions:**

See weaknesses.

**Limitations:**

the authors adequately addressed the limitations

---

> ### Author Rebuttal · Authors · 2024-08-06
>
> **W1. The dynamic information flow control does not have that much technical novelty, as it is quite close to what IBD does.**
>
> **A1.** We thank the reviewer for the valuable feedback, and we would like to clarify the primary differences between our CODE and IBD [R1].
>
> The dynamic information flow control in our method consists of two main regulating terms, $\alpha_{t}$ and $\beta_{t}$. The first term, $\alpha_{t}$, is used to adjust the penalizing original logit values in contrastive decoding. In IBD, this term is obtained by constructing another image-biased model and calculating the JSD with the vanilla model. In contrast, our method uses only self-generated image descriptions from the model itself (thus reflecting its current understanding of the visual inputs in the textual representation) to calculate BD and obtain $\alpha_{t}$. This approach eliminates the need to build a new model and allows for seamless application to other LMMs.
>
> The second term, $\beta_{t}$, is designed to dynamically control the threshold of $V_{head}$. In IBD, $\beta$ is set to a constant value of 0.1 and is not dynamically regulated. Thus, $\beta_{t}$ is our novel feature for CODE that can expand the token searching pool when the next token prediction considering distributional difference between visual contents and their comprehensive descriptions.
>
> ---
>
> **W2. Like CODE, VCD also contrasts image with modified inputs (i.e. corrupted images), but not discussed in detail in this paper. What makes contrasting image descriptions better than contrasting with corrupted images?**
>
> **A2.** We respectfully argue that we have discussed the methodological differences of our framework with VCD in detail (line 43-44, line 102-104, line 160-162, and line 200-201), because VCD is one of the important baselines for experimental comparison. We would like to highlight again that, unlike the VCD that leverages the contaminated visual inputs with Gaussian noise, our method utilizes self-generated description from models themselves as contrasting counterparts. Our design is more reasonable because the comprehensive description reflects the current model's understanding of the visual inputs and integrates this understanding into the decoding process by penalizing original logit information, whereas VCD depends on the number of noise injection steps $T$ and could suffer from unexpected adversarial effects resulting from the noise. To improve clarity, we will add this discussion to the potential final version.
>
> ---
>
> **W3. What makes a good image description for CODE to use is not discussed.**
>
> **A3.** Here, as we described in Table. 5 of Appendix. A, we carefully design the instruction for LMMs to obtain comprehensive descriptions for the given visual contents, and its curated prompt aims for the self-generated descriptions to span possible visual contents thoroughly, answering any potential questions about the given image.
> Additionally, as we analyzed in Sec. 3.1 and Fig. 2, following the amateur model selection philosophy of contrastive decoding, the generated comprehensive descriptions reflect the given visual understanding from the models themselves and subsequentially contrasts with the original logit information to enhance response coherence during the decoding phase.
>
> Although we have demonstrated the effectiveness of self-generated descriptions in our framework across 6 LMMs and 6 benchmarks, as the reviewer pointed out, there could be more optimal descriptions that can further mitigate hallucinatory responses. This could be an interesting future research direction, and we will definitely include this discussion in the potential final version.
>
> ---
>
> [R1] IBD: Alleviating Hallucinations in Large Vision-Language Models via Image-Biased Decoding, *arxiv preprint*, 2402.18476

---

> > ### Comment · Reviewer_9CKq · 2024-08-08
> >
> > The response addresses my concerns on technical novelty and comparison with baselines. However, I am still curious about the effect of CODE with different image description prompts.

---

> ### Author Response · Authors · 2024-08-10
>
> We thank the reviewer for replies and further questions. As the reviewer requested, to investigate the effectiveness of our design choice for the detailed description prompt, we compared it with a different variation, which is a base prompt to obtain a detailed description.
>
> - Base prompt: “Provide a detailed description of the image.”
> - Our prompt: “Provide a detailed description of the image, covering all visible elements and their interactions, so as to thoroughly answer any potential questions about the image."
>
> As in the table below, we have analyzed the performance of the mmHalBench and compared the average $\sharp$ token lengths for the responses generated by the models. The results show that the average token length of our curated prompt, designed to include a more comprehensive description of all visible elements in the image, demonstrates a longer token length (approx. 24\% longer) and slightly improved performance compared to the baseline. This indicates that the information amount of the comprehensive description indeed affects to the CODE by reflecting the current visual understanding of LMMs, and effectively mitigates hallucinatory responses. We will include these results and discuss how the prompt designs can boost the CODE performance in the final version.
>
> | Models |       |   LV1.5   |                 |       |  IXC2-VL  |                 |       |  IVL-1.5  |                 |
> |----------|:-----:|:---------:|:---------------:|:-----:|:---------:|:---------------:|:-----:|:---------:|:---------------:|
> | mmHal    | Overall$\uparrow$ | Hal$\downarrow$ | $\sharp$ token | Overall$\uparrow$ | Hal$\downarrow$ | $\sharp$ token | Overall$\uparrow$ | Hal$\downarrow$ | $\sharp$ token |
> | Base |  2.34 |   52.08   |      100.54     |  3.19 |   30.21   |      56.98      |  3.42 |   33.00   |      136.16     |
> | Ours     |  2.49 |   51.00   |      115.46     |  3.46 |   25.00   |      80.73      |  3.52 |   30.21   |      195.54     |

---

### Official Review · Reviewer_FLJA · 2024-07-10

**Soundness:** 3
**Presentation:** 4
**Contribution:** 3
**Rating:** 6
**Confidence:** 4

**Summary:**

The paper proposed a contrast decoding method named CODE for large multi-modal models. CODE, as mentioned by its name uses self-generated description as contrasting references during the decoding phase of LMMs to mitigate the hallucination issues. CODE works by dynamically considering the variations between the visual features and their corresponding pure language features (description) to improve response alignment with actual visual content and misalignment with the wrong part in descriptions. Based on contrastive decoding, the author proposed dynamic restriction, which regulates the information flow, and adaptive information constraint to filters out less plausible tokens constraint in the contrastive decoding phase. The proposed method is verified on 6 benchmarks.

**Strengths:**

1.	The writing logic is relatively clear and easy to understand. The figures and tables are neat, beautiful and intuitive.
2.	The proposed method is reasonable and has a good performance.
3.	The paper conducts fundamental sufficient experiments on 6 VQA benchmarks. The ablation study shows that both DC and AIC can improve the performance of MMVP and LLaVA QA 90.

**Weaknesses:**

1. The method has to generate a comprehensive description for each image before doing the visual question answering, the inference time is long and not convenient.
2. Only conduct ablation study of DC and AIC on MMVP and LLaVA QA 90, lack of ablation study on other benchmarks, such as, POPE.
3. Lack of analysis: why the performance of proposed method is not optimal or even inferior to the underlying Greedy decoding on some problem types of MMVP. For example, Color and Appearance in MMVP.

**Questions:**

1. Have the authors tried other divergence measures such as KL divergence for the Bounded Divergence used in CODE?
2. For the computational analysis in Table 4, have you considered the cost of generating the comprehensive descriptions?
3. A question about the DC, I don't understand, when $\mathcal{D}_{\mathrm{bd}}\left(P_t^v \| P_t^d\right)$, which means there is little difference between decoding based on visual features and decoding based on pure language features. It is intuitive to decode without regard to the variations, but why does the CODE highly consider the variation?

**Limitations:**

As mentioned in the first point of Weakness.

---

> ### Author Rebuttal · Authors · 2024-08-06
>
> **W1. The method has to generate $\cdots$ convenient.**
>
> **A1.** Even if we have discussed in Discussion section and computational analysis in Table. 4, we acknowledge that our method requires additional computational resources to obtain (self-generated) textual descriptions from models themselves as visual counterparts for CD. However, we would like to highlight that, by contrasting with the models’ descriptive self-understanding, our method can mitigate the hallucinatory responses without further training, which is also one of important research venues for real-world applications.
>
> ---
>
> **W2. Only conduct ablation study of DC and AIC $\cdots$ such as, POPE.**
>
> **A2.** Thank you for the valuable feedback. We would like to clarify the effectiveness of DR and AIC designs by conducting further ablation studies. In Table. 3 of our manuscript, we have selected two benchmarks for the ablation study, one each from discriminative and generative categories (MMVP, LLaVA-QA). As the reviewer asked, we experimented additional ablation studies for the rest of the benchmarks (POPE and mmHalBench). Note that we randomly sampled 500 examples from POPE and reported accuracy. For mmHalBench, we reported overall scores. As in the below table, we can observe that the use of DR and AIC progressively enhance the performance in both benchmarks, which indicates that dynamically restricting information is an important design of the CODE implementation. We will incorporate these results in the potential next version.
>
> |              |              |       | POPE |        |       | mmHal |        |
> |:--------------:|:--------------:|:-------:|:------:|:--------:|:-------:|:-------:|:--------:|
> | DR | AIC | LV1.5 | LV-N | IVL1.5 | LV1.5 | LV-N | IVL1.5 |
> | X | X | 74.4  | 84.8 | 75.8   | 1.99  | 3.09  | 3.39   |
> | X | O | 78.8  | 85.0 | 82.0   | 2.08  | 3.33  | 3.49    |
> | O | X | 82.6  | 85.0 | 84.0   | 2.02  | 3.13  | 3.40    |
> | O | O | **86.8**  | **85.6** | **84.8**   | **2.49**  | **3.43**  | **3.52**   |
>
> ---
>
> **W3. Lack of analysis: why the performance $\cdots$ in MMVP.**
>
> **A3.** We appreciate the reviewer’s comment. First of all, there is a minor typo in the IXC2-VL result (in Table. 1, “Presence of Specific Features”), which should be 6.0 $\rightarrow$ 60.0.
>
> We have added statistical results to the table below for each category and conducted an analysis of 54 results (total 6 models and 9 question categories). Especially, as in the below summarized results, CODE boost performances for the specific visual categories where vanilla greedy decoding scored very low. This is attributed to the utilization of self-generated description that can effectively contextualize and integrate nuanced information to conduct complex categories like “State and Condition” and “Structural and Physical Characteristics”.
>
> Even if our method improve overall performance with consistency across 6 models, in visual categories such as "Orientation and Direction", our method slightly underperformed compared to the greedy decoding. This may because such categories (including other categories that the improvements seemingly marginal) require straightforward decisions that are enoughly handled by greedy decoding. For example, in orient. and quantity. categories, the answer is typically deterministic (*e.g.,* left, right, 1, 2, 3), which does not require a wider token searching pool compared to other types of questions. In these instances, the use of CODE does not seem to optimally benefit simpler tasks. We will definitely add these discussions and analyses to the potential final version.
>
> |        |  Total  | orient. | feature. |  state. | quantity. | position. | color. | structure. |  text. | viewpoint. |
> |:------:|:-------:|:-------:|:--------:|:-------:|:---------:|:---------:|:------:|:----------:|:------:|:----------:|
> | greedy |  34.11  |  33.34  |   30.15  |  13.89  |   27.08   |   28.33   |  57.78 |    13.89   |  70.00 |    32.50   |
> |  ours  |  37.95  |  32.70  |   34.09  |  22.59  |   27.08   |   36.67   |  60.38 |    22.22   |  71.67 |    34.17   |
> |  $\Delta$ | +10.1\% |  -1.9\% |  +11.6\% | +38.5\% |   +0.0\%  |  +22.73\% | +4.3\% |   +37.5\%  | +2.4\% |   +4.9\%   |
>
> ---
>
> **Q1. Have the authors tried $\cdots$ used in CODE?**
>
> **A4.** Yes, our initial design was using KL-divergence to measure divergence. However, we would like to kindly remind that the most challenging point is that the vanilla KLD is not bounded (infinite upper bound), so that it is not practical to directly use of the divergence measurements. Therefore, Utilizing the Bounded Divergence (BD) that we used for CODE implementation can address such challenge by tightening the divergence range to 0~1, which is easily incorporated to DR and AIC as an adaptive hyper-parameter form.
>
> ---
>
> **Q2. For the computational analysis $\cdots$ comprehensive descriptions?**
>
> **A5.** Yes, the computational analyses in Table. 4 have considered all the computational costs of generating self-generated descriptions.
>
> ---
>
> **Q3. A question about the DC, $\cdots$ why does the CODE highly consider the variation?**
>
> **A6.** We would like to clarify the small misunderstanding. As the reviewer mentioned, if the two logit values from the visual input and its textual description are almost identical (thus minimal variation), the decoding process should indeed prioritize the original information (the reviewer’s understanding is correct). The one missing point is that our CODE process adds up the multiplication of the logit variation and the $\alpha_{t}$. Thus, in ideal case, when the logit difference nearly gets to zero, the final addition to the original logit is canceled out, even if the $\alpha_{t}$ is close to 1. Please note that this design can be universally applied without loss of generality across different logit difference scenarios.

---

> > ### Comment · Reviewer_FLJA · 2024-08-12
> >
> > Thank you for the detailed response, it addresses my concerns on some details of the method. I tend to hold the score.

---

### Official Review · Reviewer_Z6jD · 2024-07-12

**Soundness:** 2
**Presentation:** 2
**Contribution:** 2
**Rating:** 4
**Confidence:** 4

**Summary:**

Large Multi-modal Models (LMMs) have made significant strides in understanding visual context and generating coherent responses. However, they face challenges such as hallucinations, where responses are incorrect and unrelated to visual inputs. To tackle this issue, this paper proposes COuntering DEscription Contrastive Decoding (CODE). CODE uses self-generated descriptions as reference points during decoding to mitigate hallucinations. By aligning responses with visual content through dynamic adjustments in token predictions, CODE enhances coherence and informativeness without requiring additional training. Experimental results demonstrate CODE's effectiveness in reducing hallucinations and improving cross-modal consistency across various benchmarks and state-of-the-art LMMs.

**Strengths:**

1. The paper is well-written and easy to understand.
2. The use of description to enrich language information for contrastive decoding to address hallucination problems is interesting.
3. Extensive experiments  are conducted to evaluate the proposed method.

**Weaknesses:**

1. From Table 1, it can be seen that CODE does not bring much performance improvement compared to greedy decoding in practice.

2. Did the authors consider evaluating with the CHAIR metric on generative tasks?

3. Can CODE be used in conjunction with VCD or OPERA?

**Questions:**

see weaknesses.

**Limitations:**

Yes

---

> ### Author Rebuttal · Authors · 2024-08-06
>
> **W1. From Table 1, it can be seen that CODE does not bring much performance improvement compared to greedy decoding in practice.**
>
> **A1.** We respectfully argue that the use of CODE shows consistent performance improvements across 6 different models with varying sizes. Especially, when considering the performance improvements on the more challenging generative benchmarks (Table 2) and in-the-wild benchmarks (Fig. 5), which extend beyond simple multiple-choice tests, our method shows improvements exceeding 10%, as described in lines 272-277. These gains are not marginal, and importantly note that the improvements are achieved without further training of the models.
>
> ---
>
> **W2. Did the authors consider evaluating with the CHAIR metric on generative tasks?**
>
> **A2.** We thank for the reviewer’s valuable discussion point! As an initial design of our experiments, we have considered CHAIR benchmark for generative task. However, as Ben-Kish *et al.* [R1] pointed out, CHAIR is out-of-dated measurements that is limited to only 80 object annotations in the MS-COCO. As a result, we selected recently introduced GPT-aided measurements (MMHal-Bench and LLaVA-QA) for more detailed generative evaluation.
>
> However, we further conducted experiments to clarify the effectiveness of our method on CHAIR benchmark. We have randomly sampled 500 samples from COCO and reported two metric variations per-sentence ($C_{S}$) and per-instance ($C_{I}$) proportion with context length of 64 in the below table. As in the table, our method shows competent performance and consistent results than the baselines decoding methods (greedy, VCD, and OPERA) with varying size of LMMs (LLaVA-1.5, IXC2-VL, and InternVL 1.5). We will incorporate the full CHAIR results into the potential next version.
>
> | CHAIR |        |      | LLaVA-1.5 |          |        |          | IXC2-VL |          |        |      | InternVL |         |
> |:--------:|:------:|:----:|:---------:|:--------:|:------:|:--------:|:-------:|:--------:|:------:|:----:|:--------:|:-------:|
> | | greedy |  vcd |   opera   |   ours   | greedy |    vcd   |  opera  |   ours   | greedy |  vcd |   opera  |   ours  |
> |  $C_{S}$ |  26.4  | 28.6 |    25.6   | **24.8** |  26.8  |   23.8   |   24.8  | **22.0** |  18.2  | 19.2 | **17.4** |   17.6  |
> |  $C_{I}$ |  11.1  | 12.1 |    11.5   | **10.9** |   11.8  | **10.2** |   11.1  |   10.6   |  10.1  | 10.4 |   10.8   | **9.7** |
>
> ---
>
> **W3. Can CODE be used in conjunction with VCD or OPERA?**
>
> **A3.** CD-based decoding methods essentially requires a contrasting counterpart to penalize the original information with the logit variance. Our method contrasts with the information from self-generated comprehensive description, while VCD and OPERA rely on distorted visual inputs and logit penalties on pre-mature layers for contrastive decoding, respectively. Due to the different contrastive designs of each framework, conjunction of each method is not a feasible option. Additionally, simply concatenating each decoding process may result in the over-penalization of the original information, so that we cannot assure the right models' responses. Also, considering the increased inference time for each decoding framework, we believe that directly combining CD-based methods is not practical.
>
> ---
> [R1] Mitigating Open-Vocabulary Caption Hallucinations, *arxiv preprint*, 2312.03631

---

> > ### Comment · Reviewer_Z6jD · 2024-08-09
> >
> > Thanks for you repsonses! I still have some concerns.
> >
> > 1. How about the performance of the  greedy decoding in Figure 5？
> >
> > 2. From Table 1 and 2, I think CODE does not always provide much improvement compared with the greedy decoding. It seems that only LLaVA-NeXT and InternVL have a obvious  improvement. I think this is a limitation.

---

> ### Author Response · Authors · 2024-08-11
>
> We would like to appreciate to the reviewer for the active engagement and discussion. We address each question separately below.
>
> ---
>
> **A1.** As the reviewer asked, we conducted additional experiments on two in-the-wild benchmarks (RealW-QA and LLaVA(W)) and reported greedy decoding performance in the table below. Note that these results slightly underperform compared to other baseline decoding methods.
>
> |          |  LV1.5 | LV1.5 | IXC-VL | IXC-VL | IVL-1.5 | IVL-1.5 |
> |----------|-------:|------:|-------:|-------:|--------:|--------:|
> |          | greedy |  ours | greedy |  ours  |  greedy |   ours  |
> | RealW-QA      |   51.2  |  56.2 |   56.9  |  62.4  |   63.4   |   66.3  |
> | LLaVA(W) |  69.3  |  72.6 |  73.3  |  82.3  |   84.8  |   85.7  |
>
>
> ---
>
> **A2.** We acknowledge that the performance gain always cannot be significantly improved, especially considering our extensive experimental comparison across **6 benchmarks and 6 varying size of LMMs**. We will definitely integrate the pointed-out limitation into the final version.
>
> Additionally, we kindly encourage the reviewer to refer to our response for R#FLJA (A3). Based on the additional analysis on MMVP, our CODE method shows more powerful performance especially in the generative tasks that require more comprehensive understanding of the given visual context, compared to the deterministic tasks (such as multiple-choice questions), which are relatively simple and straightforward. We believe that this is attributed to our CODE strategy, which utilizes the comprehensive description from the models themselves as restricting information flow when sequentially generating response tokens, thus showing fewer hallucinatory responses during more longer context generation (beyond a single answering: yes/no or multiple-choice question).

---

### Author Rebuttal · Authors · 2024-08-06

We would like to thank the reviewers for the constructive feedback, which we will incorporate into the potential revised version. We also appreciate to all reviewers (*Z6jD*, *FLJA*, *9CKq*) for acknowledging the novelty of our paper, which is the use of self-generated comprehensive descriptions to mitigate hallucinatory responses from existing LMMs.

We have carefully reviewed several concerns that reviewers raised and responded to each question individually. Please feel free to discuss any remaining concerns that can improve our manuscripts during the discussion period.

---

### Author Response · Authors · 2024-08-12

We sincerely appreciate for the acitive discussion and responses for the reviewers during the discussion period. We have carefully revisited additional questions and addressed them comprehensively. We respectfully ask reviewers to review our responses, also happy to address any remaining concerns. We are eager to integrate all your feedbacks into our final version.

---

### Author Response · Authors · 2024-08-13

Dear reviewers,

As the discussion period will end soon, we kindly ask that you review our additional responses, in which we have made every effort to comprehensively address your remaining questions.

If you have any further concerns, we would greatly appreciate it if you could share them with us, and we would be happy to discuss them with you.

---

### Decision · Program_Chairs · 2024-09-25

**Decision:**

Accept (poster)

**Comment:**

This paper proposed a contrast decoding method called CODE for handling hallucinations of large multi-modal models. It is interesting to use descriptions for enriching language information to address hallucination problems. The idea of contrast image and caption in the VLM decoding process is new, and the proposed method is quite effective on multiple benchmarks. Due to these merits, most of the reviewers are positive about this paper. Though one of the reviewer still has concerns about the details of experiments, authors have addressed them during rebuttal. Thus, AC recommends acceptance.